# MicroRNA-9-5p-CDX2 Axis: A Useful Prognostic Biomarker for Patients with Stage II/III Colorectal Cancer

**DOI:** 10.3390/cancers11121891

**Published:** 2019-11-27

**Authors:** Aya Nishiuchi, Shigeo Hisamori, Masazumi Sakaguchi, Keita Fukuyama, Nobuaki Hoshino, Yoshiro Itatani, Shusaku Honma, Hisatsugu Maekawa, Tatsuto Nishigori, Shigeru Tsunoda, Kazutaka Obama, Hiroyuki Miyoshi, Yohei Shimono, M. Mark Taketo, Yoshiharu Sakai

**Affiliations:** 1Department of Surgery, Graduate School of Medicine, Kyoto University, Kyoto 606-8507, Japan; annya@kuhp.kyoto-u.ac.jp (A.N.); maskgch@kuhp.kyoto-u.ac.jp (M.S.); hoshinob@kuhp.kyoto-u.ac.jp (N.H.); itatani@kuhp.kyoto-u.ac.jp (Y.I.); shomma74@kuhp.kyoto-u.ac.jp (S.H.); hisatsug@kuhp.kyoto-u.ac.jp (H.M.); nsgr@kuhp.kyoto-u.ac.jp (T.N.); tsunoda@kuhp.kyoto-u.ac.jp (S.T.); kobama@kuhp.kyoto-u.ac.jp (K.O.); ysakai@kuhp.kyoto-u.ac.jp (Y.S.); 2Department of Gastroenterological Surgery, Osaka Red Cross Hospital, Osaka 543-8555, Japan; 3Department of Clinical Oncology, Kyoto University Hospital, Kyoto 606-8507, Japan; fkeita@kuhp.kyoto-u.ac.jp; 4Division of Experimental Therapeutics, Department of Gastrointestinal Surgery, Graduate School of Medicine, Kyoto University, Kyoto 606-8501, Japan; miyoshi.hiroyuki.5w@kyoto-u.ac.jp; 5Department of Biochemistry, School of Medicine, Fujita Health University, Aichi 470-1192, Japan; yshimono@fujita-hu.ac.jp; 6Division of Experimental Therapeutics, Graduate School of Medicine, Kyoto University, Kyoto 606-8501, Japan; taketo@mfour.med.kyoto-u.ac.jp

**Keywords:** CDX2, stage II/III colorectal cancer, microRNA-9-5p

## Abstract

A lack of caudal-type homeobox transcription factor 2 (CDX2) protein expression has been proposed as a prognostic biomarker for colorectal cancer (CRC). However, the relationship between CDX2 levels and the survival of patients with stage II/III CRC along with the relationship between microRNAs (miRs) and CDX2 expression are unclear. Tissue samples were collected from patients with stage II/III CRC surgically treated at Kyoto University Hospital. CDX2 expression was semi-quantitatively evaluated by immunohistochemistry (IHC). The prognostic impacts of CDX2 expression on overall survival (OS) and relapse-free survival (RFS) were evaluated by multivariable statistical analysis. The expression of miRs regulating CDX2 expression and their prognostic impacts were analyzed using The Cancer Genome Atlas Program for CRC (TCGA-CRC). Eleven of 174 CRC tissues lacked CDX2 expression. The five-year OS and RFS rates of patients with CDX2-negative CRC were significantly lower than those of CDX2-positive patients. Multivariate analysis of clinicopathological features revealed that CDX2-negative status is an independent marker of poor prognosis in stage II/III CRC. miR-9-5p was shown to regulate CDX2 expression. TCGA-CRC analysis showed that high miR-9-5p expression was significantly associated with poor patient prognosis in stage II/III CRC. In conclusion, CDX2, the post-transcriptional target of microRNA-9-5p, is a useful prognostic biomarker in patients with stage II/III CRC.

## 1. Introduction

Colorectal cancer (CRC) is one of the leading causes of cancer-related deaths worldwide [1], and great efforts have been made to develop more effective treatment strategies, leading to improvements in the prognosis of patients with CRC. Surgery is the main treatment for patients with pathological stage II CRC, whereas patients with stage III CRC are treated surgically followed by adjuvant chemotherapy [2]. Previous reports described a “high-risk stage II CRC” subgroup which present several known risk factors of recurrence, including the pathological T4 status, suboptimal lymph node retrieval, bowel obstruction or perforation, and lymphatic and venous invasion [3,4,5]. However, the prognostic impact of these risk factors and benefits of adjuvant chemotherapy in these subgroups remain controversial [6]. While oxaliplatin and fluoropyrimidine-based chemotherapies are the standard adjuvant treatments for stage III CRC [7,8,9], these chemotherapy regimens often have little effect and can be severely toxic to some patients. Thus, a simple and reliable biomarker, useful for identifying subgroups of patients with aggressive stage II/III CRC, is needed as these patients are likely to benefit from adjuvant chemotherapy.

Recent studies suggested caudal-type homeobox transcription factor 2 (CDX2) as a new prognostic biomarker in patients with CRC [10,11]. *CDX2* functions as a tumor suppressor gene that maintains the intestinal epithelium, adhesion, proliferation, and apoptosis [12,13,14] and its expression levels are reduced in most human colon cancer tissues [15]. Interestingly the other report proposed that lack of CDX2 was associated with low E-cadherin expression, tight junction disruption and epithelial-to-mesenchymal transition independently of tumor budding [16]

Genetic alterations in the *CDX2* locus are rarely found in CRC [17,18,19], and therefore epigenetic modifications of *CDX2* may be a main driving force in CRC progression. MicroRNAs (miRs) are small non-coding regulatory RNAs that generally negatively modulate translation through complementary binding to the 3′ untranslated region (3′-UTR) of their target mRNAs [20]. In CRC, numerous miRs have been reported as being either tumorigenic or tumor-suppressive, and are often correlated with prognosis [21,22,23]. We hypothesized that miRs are associated with post-transcriptional gene silencing of *CDX2* in CRC. 

This study was performed to evaluate the validity of CDX2 as a prognostic factor in patients with stage II/III CRC by using clinical tumor samples and to explore the specific miRNAs targeting CDX2. We found that the prognosis of the CDX2-negative group was significantly worse than that of the CDX2-positive group in our cohort of patients with stage II/III CRC. Moreover, we found that miRNA-9-5p directly suppresses CDX2 expression at the post-transcriptional level and affects the prognosis of patients with CRC in an opposite manner to the expression of CDX2.

## 2. Results

### 2.1. Lack of CDX2 Expression is Associated with Poor Prognosis of Stage II/ III CRC 

Initially, a total of 185 patients were enrolled; of these, 11 patients were excluded, as shown in Figure 1a. Eleven (6.3%) of the 174 patients with CRC lacked CDX2 expression. This CDX2-negative group consisted of two staining patterns; a score of 0 (a complete loss of CDX2 expression) was observed in 1.7% of patients (*n* = 3/174) (Panel A), and a score of 0.5 (scattered and faint CDX2 expression in a minority of tumor cells) was found in 4.6% of patients (*n* = 8/174) (Panel B). The CDX2-positive group showed two staining patterns: a score of 2 (moderate/strong staining in most tumor cells) was observed in 37.9% of patients (*n* = 66/174) (Panel C), and a score of 3 (strong staining in all tumor cells) was observed in 55.7% of patients (*n* = 97/174) (Panel D) (Figure 1b).

The clinicopathological characteristics of patients separated into CDX2-positive and CDX2-negative groups are shown in Table 1. In the CDX2-negative group, the rate of disease in the right colon and Por/Sig/Muc histology type were significantly higher than in the CDX2-positive group (both *p* < 0.001). The results of univariate and multivariate analyses for relapse-free survival (RFS) are shown in Table 2. Both the expression levels and Por/Sig/Muc histology in the CDX2-negative group were significantly associated with a lower RFS; furthermore, in multivariate analysis, CDX2-negative status was found to be an independent factor for poor prognosis (hazard ratio 4.33; 95% CI, 1.37–12.3; *p* = 0.014).

Kaplan–Meier survival analysis revealed that the CDX2-negative group was significantly associated with a lower five-year overall survival (OS) and lower five-year RFS (both were *p* < 0.001, Figure 1c,d). The median follow-up periods for OS and RFS were 59.0 months (range, 6.2–79.3 months) and 58.6 months (range, 6.2–77 months), respectively.

### 2.2. MiR-9-5p was Upregulated in CDX2^low^ Tumor Samples

Following fixation and permeabilization, epithelial cell adhesion molecule EpCAM^+^/FVD^−^/CDX2^high^ and EpCAM^+^/FVD^−^/CDX2^low^ tumor cells were double-sorted, and total RNA was extracted from two patient-derived xenografts (PDXs) and one spheroid sample (Figure 2a). Based on computational prediction, eight miRs (miR-9-5p, miR-15a-5p, miR-16-5p, miR-22-3p, miR-24-3p, miR-195-5p, miR-204, and miR-211-5p) were selected as candidates that downregulate CDX2 expression (Figure 2b). Among these, only miR-9-5p was significantly overexpressed in the CDX2^low^ tumor population as compared to in the CDX2^high^ population in the three samples (all three samples, *p* < 0.05, Figure 2c). miR-195-5p was detected in only one sample (data not shown) and miR-211-5p was undetectable.

### 2.3. MiR-9-5p Targets CDX2

A PCR product representing the *CDX*2 3′-UTR and containing the target site for miR-9-5p was inserted downstream of the luciferase minigene in the pGL3-control vector (Figure 3a). SW480 cells were then co-transfected with the pGL3-control vector, phRL-TK Renilla luciferase vector, and miR-9-5p precursor or its negative control. We found that co-transfection of the miR-9-5p precursor suppressed the normalized luciferase activity of the vectors but that a mutation in the miR-9-5p precursor abolished this inhibition (Figure 3b).

In addition, SW480 cells were found by RT-PCR to express *CDX2* (Figure 3c); western blotting showed that CDX2 protein levels decreased when co-transfection was performed with the miR-9-5p precursor (Figure 3d). 

### 2.4. Overexpression of miR-9-1 Accelerates Cell Cycle and Cell Proliferation and Increases Sensitivity to Anti-Cancer Drugs 

The mature miR-9 transcript is produced by three independent genes: *miR-9-1*, *miR-9-2*, and *miR-9-3*. Of miR-9-5p and miR-9-3p which are mature miR-9s, miR-9-5p is dominant in humans [24] (Figure 4a). We performed cell proliferation, cell cycle, and cytotoxicity assays using DLD-1 cells overexpressing miR-9-1 (DLD-1-miR-9-1) which stably expresses miR-9-5p (Figure 4b). CDX2 expression was decreased by both transient miR-9-5p expression and stable miR-9-1 expression (Figure 4c). 

A significant increase in cell proliferation was observed in DLD-1-miR-9-1 as compared to in the control (Figure 5a). Insertion of miR-9-1 caused a significant decrease in the G0/G1 population and corresponding increase in the S and G2/M populations (Figure 5b). Overexpression of miR-9-1 significantly increased the sensitivity to two of three anti-cancer drugs, 5-fluorouracil (Wako, Osaka, Japan), oxaliplatin (L-OHP; Wako) or irinotecan (CPT-11; Wako), which are frequently used for clinical treatment of CRC (Figure 5c).

Next, we investigated the relationship between the expression of CDX2 and pathological therapeutic effect of neoadjuvant chemotherapy (NAC)/neoadjuvant chemoradiotherapy (NACRT) followed by surgical resection. A therapeutic effect of Grade 2 was observed in one of the three CDX2-negative cases, whereas all eight CDX2-positive cases showed a poorly response to NAC/NACRT (Figure 5d).

### 2.5. MiR-9-5p Expression in Patients with CRC Correlates with Poor OS 

We analyzed the relationship between miR-9-5p expression levels and prognosis using TCGA database. The OS of the miR-9-5p high-expression group was significantly shorter than that of the miR-9-5p low-expression group in patients with stage II/III and stage I–IV CRC (*p* = 0.019 and *p* = 0.010, respectively Figure 6).

## 3. Discussion

Increasing evidence has suggested that CDX2 can be used as a new prognostic biomarker, particularly in patients with advanced CRC [14,25]. In this study, we demonstrated that low levels of CDX2 expression are an independent factor in the poor prognosis of patients with stage II/III CRC, indicating that these subgroups can greatly benefit from adjuvant chemotherapy following surgery. Moreover, we found that miR-9-5p down-regulates CDX2 expression at the post-transcriptional level in CRC. This mechanism may be a new therapeutic strategy for patients with CRC.

We examined the expression of CDX2 in 174 patients with CRC by IHC based on a previously described scoring system [25]. The CDX2-negative rate in our study was 6.3% (11/174), which was comparable to that in the same report [25]. As determined from Kaplan–Meier survival analysis, both the five-year OS and RFS were significantly lower in CDX2-negative patients. Although previous data suggested that CDX2-negative CRC is associated with either OS or DFS [25,26], this is the first study to report the relationship between CDX2 expression and both OS and RFS in CRC. In addition, the rate CDX2-negative samples were significantly higher in right colon cancer and in patients with CRC and Por/Sig/Muc histology and confirms the results of several recent reports [26,27,28,29]. 

Because of its simplicity and ease, IHC examination of CDX2 levels is widely applied in clinical studies as a sensitive marker of adenocarcinomas of intestinal origin; thus, we examined CDX2 expression in this study. Because CDX2 is expressed in the nuclei of intestinal epithelial cells [12], both fixation and membrane permeabilization are necessary for intracellular staining. In this study, we performed RT-qPCR to determine if the miRs present in the RNA-induced silencing complex was stable and therefore could be detected even in fixed and permeabilized tumor cells. This technique is innovative and helped identify which miRs are expressed in primary CRC samples. To explore this further, we first extracted total RNA from PDXs to obtain a greater amount of sample. However, using PDXs required a prolonged time to establish the primary PDXs and resulted in a low yield of viable cells that can be used for analysis. To confirm the data that obtained in PDX analysis, we additionally developed a spheroid model, which increased cell viability (data not shown). To examine whether the PDX- and spheroid-derived cells reflected the characteristics of the primary tumor, we also performed IHC examination and confirmed that the staining patterns were similar to the resected primary tumor (Appendix A). 

The results of the luciferase reporter assay and western blotting analysis suggested that miR-9-5p directly targets the *CDX2* mRNA and suppresses the protein level of CDX2. MiR-9, which is selectively expressed in neural tissues under normal conditions and regulates their development [30], behaves as an oncomiR in some cancers originating outside the nervous system, such as Hodgkin lymphomas [31], breast cancers [32], cervical cancers [33], colon cancers [34], and stomach cancers [35]. According to the report, miR-9 targets *CDH1*, the E-cadherin-encoding mRNA, which is related to increased cell motility and invasiveness and metastasis in breast cancers [32]. MiR-9 has been shown to affect various tumorigenic processes, including cellular proliferation [33,35], migration [32,34], and inflammation [31]. In support of our data, miR-9 has also been reported as a regulator of CDX2 expression in gastric cancer, where CDX2 expression may be regulated through other mechanisms such as the oncogenic *ras*-activated PKC pathway or *CDX2* mutations [35]. Our in vitro results were consistent with those observed in previously reported phenotypes regarding decreased expression of CDX2. Furthermore, from TCGA analyses, we observed a consistent trend: OS in the miR-9-5p high-expression group was worse than that in the miR-9-5p low-expression group.

There were several limitations to this study. This was a single-center study, and the number of patients in the CDX2-negative group of IHC was too small for robust statistical analysis. In addition, the number of tumor samples used in fluorescence-activated cell sorting (FACS) analysis was small. However, the result from the three samples showed comparable patterns, suggesting that the data are reliable. Chemotherapy has been reported to be more effective in CDX2-negative CRC, but our results did not demonstrate the benefit of chemotherapy because the number of cases was small. However, chemotherapy in some CDX2-negative CRC cases was much more effective than in CDX2-positive CRC cases.

Recent studies showed that CDX2 targets the multidrug resistance gene *MDR1* (*ABCB1*) and *CFTR* (*ABCC7*), coding for two ATP-dependent drug efflet pumps [36,37,38]. These genes are significantly down-regulated in CDX2-negative cell lines and CDX2-negative patient tumors and have also been reported to promote the response to chemotherapy [24,38]. This supports our results showing that CDX2-positive cases show a poor response to chemotherapy which may be effective in patients with CDX2-negative CRC.

## 4. Materials and Methods

### 4.1. Patient Population

We enrolled 185 patients diagnosed with stage II/III CRC who underwent complete resection of their primary tumor at Kyoto University Hospital between November 2007 and December 2010. The tumor samples were retrospectively analyzed. This study protocol was approved by the institutional review board of Kyoto University (R1251), and the patients provided their informed consent for data analysis. 

### 4.2. Immunohistochemistry (IHC)

Formalin-fixed, paraffin-embedded human colorectal tissues were stained with an anti-CDX2 mouse monoclonal antibody (Clone: CDX2-88, 1:50, BioGenex, San Ramon, CA, USA) using the avidin-biotin immunoperoxidase method. A scoring system described in a previous report [24] was used to quantify CDX2 expression levels. All IHC samples were evaluated independently by two researchers. The pathological therapeutic effects of a preoperative treatment are defined in the Japanese Classification of Colorectal Carcinoma, 8th Edition [39].

### 4.3. Data Analysis from TCGA-Colorectal Cancer (TCGA-CRC)

Data from The Cancer Genome Atlas Program (TCGA) related to CRC (COAD and READ) were obtained from the Genomic Data Commons Data Portal (https://portal.gdc.cancer.gov/). We calculated mature miR expression levels as performed by OncoLnc (http://www.oncolnc.org/) using Python scripts for processing isoform quantification data. We then integrated the clinical data and miR expression levels from the primary tumors. Patients without miR-Seq data or with a follow-up time equal to zero were excluded. The cut-off value for survival analysis of miR-9-5p was determined to be 10%, which is consistent with the CDX2-negative rate reported previously and our IHC results [11,25]. R [40] and the party [41], stringr [42], survival [43], and tidyverse [44] packages were used for data processing.

### 4.4. Statistical Analysis

All statistical analyses were performed using JMP Pro software, version 12.0.0 (SAS Institute, Inc., Cary, NC, USA). Clinicopathological factors were compared using the χ^2^ test. Survival analyses were performed using the Kaplan–Meier method and examined using a log-rank test. Multivariate survival analyses were performed with Cox’s proportional hazard model. All predictors with a *p*-value of <0.05 in univariate analysis along with clinically important factors were entered into multivariate analysis. All analyses were two-sided, and statistical significance was set to a *p*-value of <0.05.

### 4.5. Cell Lines and Cell Culture

The human colon cancer cell lines, SW480 and DLD-1, were purchased from American Type Culture Collection (Manassas, VA, USA). Both cell lines were cultured in DMEM (Nacalai Tesque, Kyoto, Japan) containing 10% FBS (Life Technologies, Carlsbad, CA, USA) supplemented with a 1% penicillin/streptomycin mixture. A lentiviral GFP plasmid expressing miR-9-1 and the scrambled control plasmid were purchased from System Biosciences (Palo Alto, CA, USA). These plasmids were transfected with virus particle vectors (psPAX2 and pMD2.G) to produce recombinant lentiviruses using Lipofectamine 2000 (Thermo Fisher Scientific, Waltham, MA, USA). DLD-1 cells were infected with the lentiviruses, and DLD-1-miR-9-1 was selected using a FACS Aria II cell sorter (Becton, Dickinson and Company (BD), Franklin Lakes, NJ, USA) to collect GFP-positive cells. 

### 4.6. Patient-Derived Tumor Xenografts (PDX) and Patient-Derived Cancer-Spheroids

Primary CRC specimens were obtained from patients who underwent colorectal resection and small pieces of tissue were transplanted into the subcutaneous tissues of immunodeficient female nude mouse (KSN/Slc mouse, SLC, Shizuoka, Japan) to establish PDXs. Spheroids were cultured as previously described [45]. These experiments were approved by the institutional Animal Ethics and Research Committee, Kyoto University (MedKyo 18148).

### 4.7. Flow Cytometry

PDXs were enzymatically dissociated into a single-cell suspension using a Human Tumor Dissociation Kit (Miltenyi Biotec, Bergisch Gladbach, Germany). Single-cell suspensions from PDXs and spheroids were stained with Fixable Viability Dye (FVD, eBioscience, San Diego, CA, USA) and a mouse anti-human EpCAM antibody (Clone: EBA-1, BD). After fixation with 4% paraformaldehyde, blocking and permeabilization were performed with 1% bovine serum albumin, 5% normal goat serum, and 0.1% Triton X-100. After washing, the tissues were stained with a rabbit anti-CDX2 monoclonal antibody (Clone: EPR2764Y, diluted 1:50, Abcam, Cambridge, UK). The samples were then analyzed using a FACS Aria II cell sorter (BD). Total RNA was isolated from the CDX2^high^/CDX2^low^ cancer cells using a miRNeasy Micro Kit (Qiagen, Hilden, Germany) and converted to cDNA using a Universal cDNA Synthesis Kit II (Exiqon, Vedbaek, Denmark).

### 4.8. In-Silico Prediction of MiRs Targeting the CDX2 3′-UTR

Three independent algorithms, TargetScan (http://www.targetscan.org/), Pictar (http://pictar.mdc-berlin.de/), and MicroCosm Targ (http://www.ebi.ac.uk/enright-srv/microcosm/htdocs/targets), were used to predict the potential miRs capable of targeting the 3′-UTR of *CDX2*. To reduce the rate of false-positives, miRs predicted by at least two algorithms were validated by RT-qPCR.

### 4.9. Reverse-Transcriptase Quantitative PCR (RT-qPCR)

RT-qPCR was performed using ExiLENT SYBR Green master mix (Exiqon) on a 7900HT Fast Real-Time PCR system (Applied Biosystems, Foster City, CA, USA). Relative miR expression levels were determined using the ΔΔCt method and normalized to miR-103a-3p levels [46]. A schematic representation of the workflow is shown in Appendix A.

### 4.10. Plasmid Vectors for Luciferase Reporter Assays and Mutagenesis

A 1013-bp region of *CDX2* 3′-UTR (corresponding to positions 1278–2291 of NM_001265), containing the binding sites for miR-9-5p, was amplified by PCR using cDNA derived from DLD-1 cells as a template. The PCR product was inserted into the pGEM-T vector (Promega, Madison, WI, USA). The *CDX2* 3′-UTR product was cloned into the 3′-end of luciferase reporter vector pGL3-MC [47]. Mutation of the putative miR-9-5p target sequences within the 3′-UTR of *CDX2* was performed using a QuikChange Lightning Multi Site-Directed Mutagenesis Kit (Agilent Technologies, Santa Clara, CA, USA). Sequences of all PCR products were checked for accuracy. The Appendix A contain further details of the experiments listed there.

### 4.11. Luciferase Reporter Assay

SW480 cells were seeded at a density of 1 × 10^5^ cells per well in a 48-well plate on the day before transfection. Cells were transfected with 320 ng of the pGL3 luciferase expression construct containing either the wild-type (WT) or mutated 3′-UTR of human *CDX2* (mut), 6.4 ng of phRL-TK Renilla luciferase vector (Promega) and 25 nmol/L of the has-miR-9-5p precursor or its negative control (Life Technologies), along with Lipofectamine LTX (Life Technologies). At 48 h after transfection, luciferase activities were measured using a Dual-Luciferase Reporter Assay System (Promega) and normalized to Renilla luciferase activity.

### 4.12. Western Blotting 

SW480 cells were transfected with 25 nmol/L of the miR-9-5p precursor or its negative control (N.C.) and cultured for 2 days. Transient miR-9-5p-expressing SW480 and DLD-1 cells and stable miR-9-1-expressing DLD-1 were lysed in sodium dodecyl sulfate (SDS) lysis buffer. Proteins in the cell samples were separated by 10% SDS-PAGE, followed by transfer to a polyvinylidene difluoride membrane (Merck Millipore, Burlington, MA, USA). Immunoblotting with a rabbit anti-human CDX2 antibody (D11D10, 1:1000, Cell Signaling Technology, Danvers, MA, USA) was performed, followed by incubation with a horseradish peroxidase-conjugated secondary antibody (Dako, Glostrup, Denmark). Horseradish peroxidase-conjugated ACTB antibody (1:8000 dilution, clone AC-15, Sigma-Aldrich, St. Louis, MO, USA) was used as a loading control. A LAS-3000 mini system (Fuji Film, Tokyo, Japan) was used to measure the chemiluminescence signal.

### 4.13. Cell Proliferation, Cell Cycle, and Cytotoxicity Assays 

We used stable DLD-1-miR-9-1 and scramble control cells for these assays. A total of 0.5 × 10^6^ cells were seeded in triplicate into 6-cm dishes (day 0), and were harvested and counted in triplicate using an auto cell counter at days 1 and 4. Cell proliferation was calculated as relative rate, with day 0 value as 1. For the cell cycle assay, cells were dissociated with trypsin/EDTA, washed with PBS, and fixed with ice-cold 70% ethanol for 30 min. They were then re-suspended in PI/RNase Staining Buffer (BD) and incubated for 15 min at room temperature. The samples were analyzed using a FACS Aria II cell sorter. Cell cycle profiles were analyzed using the FlowJo (version 10) data analysis software (BD) to determine the percentage of cells in G0/G1, S, and G2/M phases. For cytotoxicity assay, cells (1 × 10^5^ cells/well) were seeded into 12-well cell culture plates. After overnight incubation, CPT-11, L-OHP or 5-fluorouracil was added at the indicated concentration, and the cell number was counted using an auto cell counter at 72 h after administration of anti-cancer drugs. Cell viability was calculated as the ratio of the treated/control cells.

## 5. Conclusions

In summary, we demonstrated that CDX2, a post-transcriptional target of microRNA-9-5p, is a useful prognostic biomarker in patients with stage II/III CRC. These clinical and molecular insights may contribute to developing a new strategy for treating CRC.

## Figures and Tables

**Figure 1 cancers-11-01891-f001:**
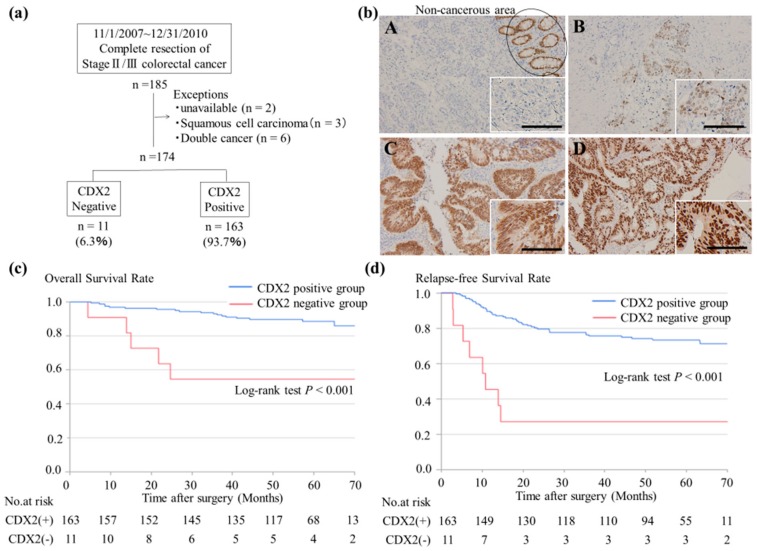
IHC examination of CDX2 in patients with stage II/III CRC: (**a**) Schematic representation of the workflow of immunohistochemistry of CDX2. (**b**) Expression of CDX2 in CRC specimens. Normal intestinal epithelial cells were used as an internal positive control. A Score 0 and B Score 0.5; were determined to be CDX2-negative. C Score 2 and D Score 3; were determined to be CDX2-positive. The scale bar represents 100 µm. (**c**,**d**) Kaplan–Meier curves for OS and RFS of the 174 patients with stage II/III CRC. CDX2: caudal-type homeobox transcription factor 2; CRC: colorectal cancer; OS: overall survival; RFS: relapse-free survival.

**Figure 2 cancers-11-01891-f002:**
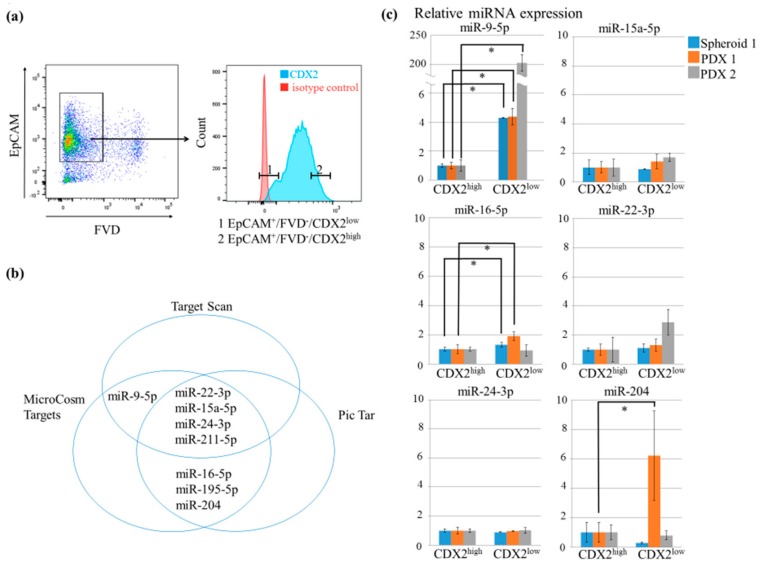
Search for miRs that regulate CDX2 expression. (**a**) Representative flow cytometry plot. EpCAM^+^/FVD^−^/CDX2^low^(1) and EpCAM^+^/FVD^−^/CDX2^high^(2) tumor cells in the same sample were collected by flow cytometry. (**b**) Computational prediction of miRNAs capable of targeting CDX2; eight miRs predicted by at least two algorithms were selected. (**c**) Comparison of the relative miRNA levels of different miRs in FACS-isolated CDX2^high^ and CDX2^low^ tumor cells. (* *p* < 0.05).

**Figure 3 cancers-11-01891-f003:**
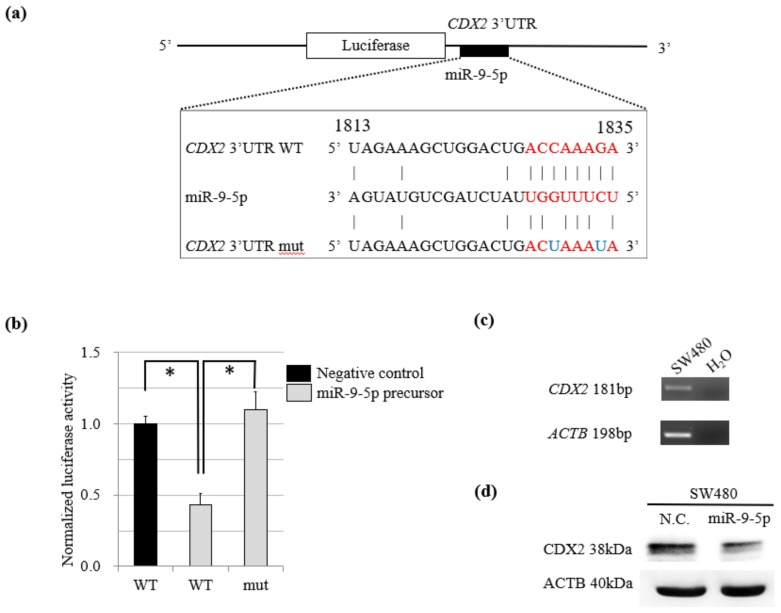
(**a**) Schematic representation of the predicted miR-9-5p target site sequence within the 3′-UTR of *CDX2*. Two nucleotides in the miR-9-5p sequence were mutated in the *CDX2* mutant plasmid. The numbers indicate the position of nucleotides in the wild-type sequence of *CDX2* (NM_001265). (**b**) The activity of the firefly luciferase gene which were inserted the *CDX*2 3′-UTR and containing the target site for miR-9-5p. The data are presented as mean and SD of separate transfections (*n* = 3, **p* < 0.05). (**c**) Semi-quantitative RT-PCR analysis of *CDX2* mRNA expression levels in SW480 cells showing the expression of *CDX2*. (**d**) Suppression of the endogenous CDX2 protein in transient miR-9-5p-expressing SW480.

**Figure 4 cancers-11-01891-f004:**
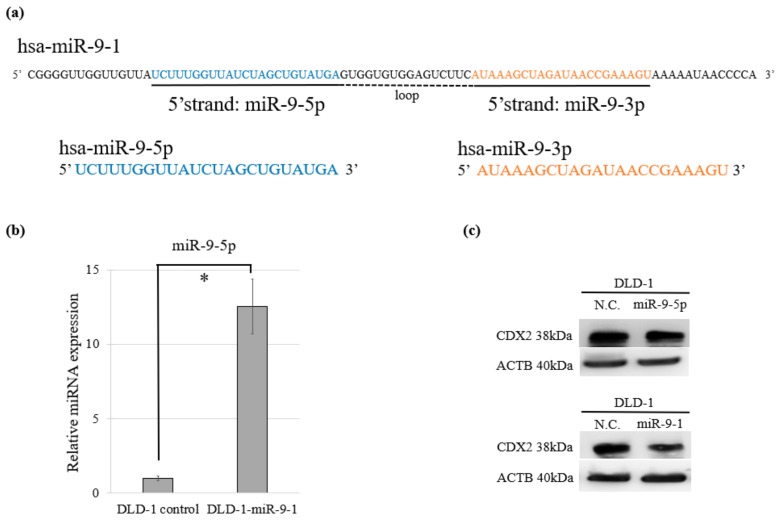
Relationship of miR-9-1 and miR-9-5p. (**a**) Each sequence of pre-miR-9-1 and its mature RNAs miR-9-5p and miR-9-3p. (**b**) Relative expression of miR-9-5p in stable DLD-1-miR-9-1 and its control. (**c**) Suppression of the endogenous CDX2 protein both in stable DLD-1-miR-9-1 and transient miR-9-5p-expressing DLD-1.

**Figure 5 cancers-11-01891-f005:**
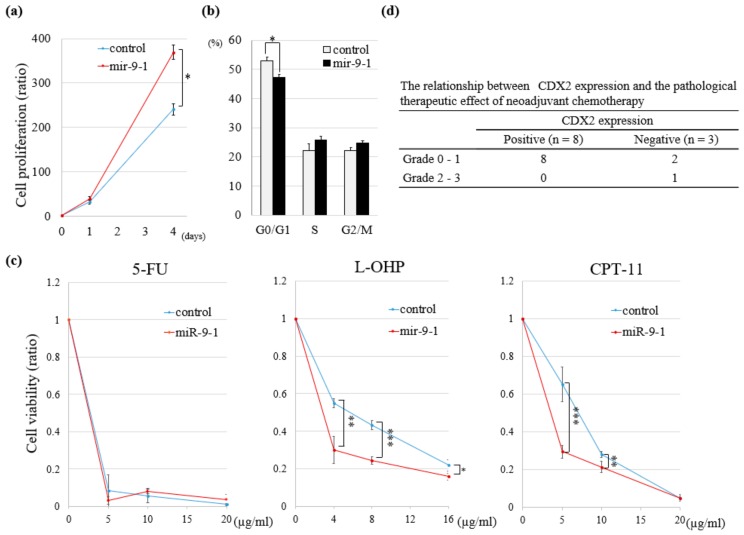
Functional assays in DLD-1-miR-9-1 and pathological therapeutic effect in NAC/NACRT cases. (**a**) Cell proliferation assay in DLD-1-miR-9-1 (*n* = 3, * *p* < 0.05). (**b**) Cell cycle assay in DLD-1-miR-9-1. Bar graph indicates the percentage of cells in G0/G1, S or G2/M phase (*n* = 3, * *p* < 0.05). (**c**) Cytotoxicity assays in DLD-1-miR-9-1. Results are presented as the mean and SD (*n* = 3, * *p* < 0.05, ** *p* < 0.01, *** *p* < 0.001). (**d**) Relationship between CDX2 expression and pathological therapeutic effect of NAC/NACRT followed by surgical resection. NAC: neoadjuvant chemotherapy; NACRT: neoadjuvant chemoradiotherapy.

**Figure 6 cancers-11-01891-f006:**
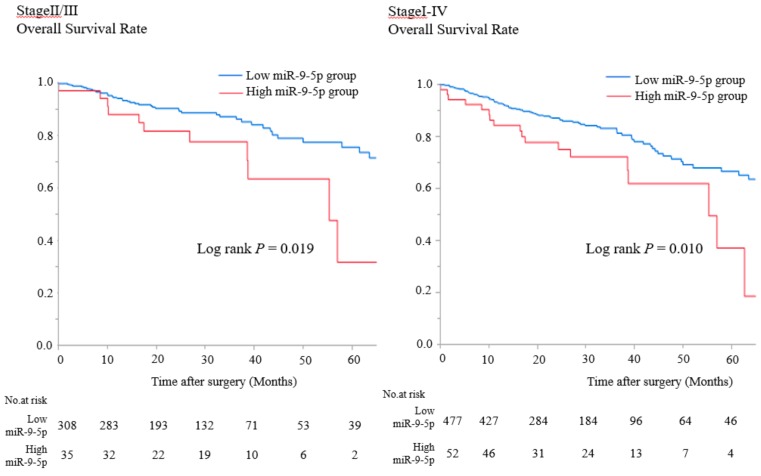
OS in the low or high miR-9-5p expressing group who underwent curative resection of stage II/III and stage I–IV CRC according to TCGA database (Kaplan–Meier estimates).

**Table 1 cancers-11-01891-t001:** Patient characteristics and CDX2 expression.

Factor		No. of Cases	CDX2 Expression		*p*-Value
			Positive (*n* = 163)	Negative (*n* = 11)	
			n (%)	n (%)	
Sex	Male	102	97 (55.8)	5 (2.9)	0.364
	Female	72	66 (37.9)	6 (3.5)	
Age (years)	<70	80	77(44.3)	3(1.7)	0.189
	≥70	94	86(49.4)	8(4.6)	
Location	Right colon	52	44 (25.3)	8 (4.6)	<0.001
	Left colon	73	73 (42.0)	0 (0.0)	
	Rectum	49	46 (26.4)	3 (1.7)	
T-factor	T 1–3	119	111 (63.8)	8 (4.6)	0.746
	T 4	55	52 (29.9)	3 (1.7)	
N-factor	N 0	95	91 (52.3)	4 (2.3)	0.201
	N +	79	72 (41.4)	7 (4.2)	
Histology	Wel/Mod	163	158 (90.8)	5 (2.9)	<0.001
	Por/Sig/Muc	11	5 (2.9)	6 (3.5)	
Lymphatic invasion	Ly 0	114	109 (62.6)	5 (2.9)	0.159
	Ly +	60	54 (31.0)	6 (3.5)	
Venous invasion	V 0	68	65 (37.4)	3 (1.7)	0.396
	V +	106	98 (56.3)	8 (4.6)	
Adjuvant chemotherapy	Yes	89	82 (47.1)	7 (4.0)	0.389
	No	85	81 (46.6)	4 (2.3)	

Well: well differentiated adenocarcinoma; Mod: moderately differentiated adenocarcinoma; Por: poorly differentiated adenocarcinoma; Sig: signet ring cell carcinoma; Muc: mucinous adenocarcinoma.

**Table 2 cancers-11-01891-t002:** Univariate and multivariate analysis for RFS.

Factor	Category	No. of Cases	Univariate Analysis			Multivariate Analysis		
			HR	95 % CI	*p-*Value	HR	95 % CI	*p*-Value
CDX2 expression	Positive	163	Ref		<0.001	Ref		0.014
	Negative	11	5.17	2.24–10.47		4.33	1.37–12.3	
Sex	Male	102	Ref		0.488	Ref		0.439
	Female	72	0.82	0.46–1.43		0.79	0.43–1.42	
Age (years)	<70	80	Ref		0.954	Ref		0.705
	≥70	94	1.02	0.59–1.77		1.12	0.70–2.33	
Location	Right colon	52	Ref			Ref		
	Left colon	73	0.91	0.47–1.81	0.784	1.37	0.65–3.08	0.416
	Rectum	49	1.07	0.53–2.19	0.850	1.60	0.71–3.75	0.260
T-factor	T 1–3	119	Ref		0.064	Ref		0.037
	T 4	55	1.70	0.96–2.95		1.87	1.04–3.30	
N-factor	N 0	95	Ref		0.091	Ref		0.109
	N +	79	1.61	0.93–2.81		1.58	0.90–2.80	
Histology	Wel/Mod	163	Ref		<0.001	Ref		0.078
	Por/Sig/Muc	11	4.95	2.15–10.05		2.61	0.89–7.01	
Lymphatic invasion	Ly 0	114	Ref		0.561			
	Ly +	60	0.84	0.45–1.49				
Venous invasion	V 0	68	Ref		0.199			
	V +	106	1.46	0.82–2.71				
Adjuvant chemotherapy	Yes	89	Ref		0.395			
	No	85	0.787	0.45–1.36				

RFS: relapse-free survival; HR: hazard ratio; CI: confidence interval; Well: well differentiated adenocarcinoma; Mod: moderately differentiated adenocarcinoma; Por: poorly differentiated adenocarcinoma; Sig: signet ring cell carcinoma; Muc: mucinous adenocarcinoma.

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
