# Peer review of "MicroRNA-9-5p-CDX2 Axis: A Useful Prognostic Biomarker for Patients with Stage II/III Colorectal Cancer"

_cancers, 2019, doi:10.3390/cancers11121891_

Round 1

Reviewer 1 Report

Molecular biomarkers assume a critical role in the characterization of colorectal cancer and in the determination of the most appropriate therapy.

During this work the Authors investigate the role of caudal-type homeobox transcription factor 2 (CDX2) protein expression as a prognostic biomarker for colorectal cancer (CRC). They also studied the relationship between CDX2 expression, survival of patients with stage II/III CRC and microRNAs (miRs). They demonstrated that CDX2-negative status was an independent marker of poor prognosis in stage II/III CRC patients. miR-9-5p was shown to regulate CDX2 gene expression. A statistical analysis showed that high expression levels of miR-9-5p were significantly associated with poor prognosis in stage II/III CRC patients.

In my opinion ths topic is interesting, the experimental design was well conducted, the results obtined support the conclutions and the paper is clearly written.

Quite all main published work related to this topic are cited in the introduction or discussion.

I only suggest the Autors to better introduce mir9 family in the introduction and discussion, so that the paper could become more fluent.

I recommend the Authors to carefully read the paper to revise some language misspellings.

Best regards

Marina De Rosa

Author Response

We appreciate for your useful suggestions. We have reviewed and added several reports describing the mechanisms of miR-9 in other cancers in the “Discussion” section L217-223.

Reviewer 2 Report

As described in manuscript, this study is a single-center study, which is a limitation. In addition, the number of CDX2-negative patients are too few for a robust statistical analysis. The data from TCGA-COAD/READ should be analyzed to support the conclusion of this study because the authors already examined the role of miR-9-5p using the TCGA-COAD/READ (Figure 3c). It is also unclear why the authors switched to TCGA-COAD/READ data, but not their own data, for miR-9-5p (Figure 3c). Lines 224-233: “The cut-off value for the survival analysis of miR-9-5p was determined to be 10% consistent with the CDX2 negative rate from previous reports and our IHC result”. However, the authors did not show the low/negative expression of CDX2 in miR-9-5p-high group, which should be clarified. Figure 3d: why only 11 patients were used for this analysis? The ratio (3/8, 37.5%) of CDX2-negative to CDX2-positive cases is not reasonable because there are only 6.3% of CDX2-negative patients (Figure 1a). The expression of miR-9-5p and CDX2 in stable DLD-1-miR-9-1 and the control cells should be confirmed. Figure 3c: the label for DLD-1-miR-9-1 and the concentration units for L-OHP and CPT-11 are missing. In addition, the exact drug dosages are not clear in this figure. In addition, the dosages of 5-FU and CPT-11 are too high for this experiment. Lower concentrations should be examined. Lines 189-191: “To examine whether the PDX- and spheroid-derived cells reflected the characteristics of the primary tumor, we also performed an IHC examination and confirmed that they showed similar staining patterns to the resected primary tumor.” This result should be shown as supplementary materials. The rationale for EpCAM/FVD staining and sorting should be explained. Lines 124-133: the format of figure legends should be corrected. The function of CDX2 should be introduced.

Author Response

Response: We appreciate your useful, insightful comments, which have helped us to significantly improve the paper. We have provided a point-by-point response to each of your comments.

Response 1) In the TCGA-COAD/READ database, we could not obtain the data for protein levels, but we obtained data on the mRNA level of CDX2. The relationship between the mRNA level of CDX2 and prognosis was examined using TCGA database, but no significant correlation was observed (Attached please find the PPT file, Ref data). However, as we described in the “Introduction” section, genetic alterations in the CDX2 gene locus are rarely found in colorectal cancer (CRC) tissue; therefore, epigenetic modifications of CDX2 may be a main driving force in suppressing the protein expression of CDX2 in CRC. The presence of CDX2 mRNA including the 3′UTR does not reflect the actual expression of CDX2 in the tissue from the perspective of microRNA functions. To avoid confusion, we did not show the CDX2-mRNA expression data with TCGA-COAD/READ in this manuscript.

Response 2) We attempted to analyze the expression of miRNA using the same cohort of patients whose pathological samples were used for CDX2 expression. However, only paraffin sections were available for this cohort, and analyzing miRNA expression using paraffin sections is difficult. Thus, we used TCGA database to investigate the relationship between miR-9-5p expression and prognosis.

Response 3) As we described above, we could not obtain the miR-9-5p values from our own dataset. We demonstrated the relationship between miR-9-5p and CDX2 in basic experiments using colon cancer cell lines. Moreover, the cut-off value for survival analysis of miR-9-5p (Figure 3e) was determined to be 10%, which is consistent with the CDX2-negative rate previously reported and in our IHC results; thus, this value was not arbitrary. Figure 3e shows that the OS was affected by the expression level of miR-9-5p.

Response 4) Eleven cases were not arbitrary extractions, but all NAC cases in our cohort. Although the number of patients was too small to examine the relationship between the effects of the NAC and CDX2 expression patterns, we showed that all eight CDX2-positive cases had poor chemotherapy outcomes. This indicates the chemo-resistance of CDX2-positive CRC.

Response 5) We have confirmed the expression of miR-9-5p in the stable DLD-1-miR-9-1 cell line. Furthermore, we confirmed that CDX2 expression level was decreased by overexpression of mR-9-1. We have added this result as a new figure in Figure 4b and 4c (L148-151) respectively.

Response 6) We apologize for showing incomplete figures. We have included the concentration units for L-OHP and CPT-11 and exact drug dosage in Figure 5c. In addition, as recommended, CPT-11 and 5-FU were tested at low concentrations. Although there was no significant difference in 5-FU, CPT-11 was significantly effective in DLD-1 overexpressing miR-9-1. These results have been added to Figure 5c (L160-163).

Response 7) We have added a representative figure of IHC examination of the primary tumor, PDX, and spheroid as Figure S1.

Response 8) We sorted EpCAM-positive and FVD-negative cells according to previous reports because “EpCAM-positive” indicates epithelial cells and FVD-negative indicates viable cells before fixation and permeabilization.

Response 9) We apologize for this error. We have corrected the format of the figure legends as you suggested.

Response 10) According to your suggestion, we have explained the function of CDX2 in the “Introduction” section (L52-57).

Reviewer 3 Report

In this article Aya Nishiuchi et al. described the relationship between CDX2 expression and miR-9-5p in stage II/III colorectal cancer. Overall the experiments are well conducted and the results are very well presented. The manuscript may be published in Cancers provided that the authors answer some questions and perform additional experiments. Indeed, the authors do not clearly indicate the interrelationship between miR-9-5p precursor and and miR-9-1. Is miR-9-5p precursor the precursor of miR-9-1? The authors must understand that a wide range of reader are susceptible to read their article and most of them are not expert of microRNA.

According to this, it is not clear why the authors used sw480 cells with miR-9-5p and DLD1 cells with miR-9-1. In figure 2C, miR-9-1 must be transfected in sw480 cells to check the activity of the reporter. Figure 2d, SW480 cells must be transfected with miR-9-1 and DLD1 with miR-9-5p to check the level of CDX2. Moreover, the effect of miR-9-5p on proliferation of SW480 must be analyzed. Only then can the authors continue their work with the DLD1- miR-9-1 cells in Figure 3.

Author Response

Response: We apologize for the confusing figures and appreciate your suggestions. The SW480 cell line was used for both experiments with transient expression of miR-9-5p and overexpression of miR-9-1. We observed that the expression of CDX2 luciferase activity was decreased in stable SW480-miR-9-1 cells, but this result was not representative. A representative figure of transient expression of miR-9-5p was presented in the manuscript. The results of western blotting of SW480-miR-9-1 cells showed that the expression of CDX2 tended to be suppressed in SW480-miR-9-1 cells (attached please find the PPT slide titled Western Blotting).

DLD-1 was also transfected transiently with miR-9-5p and the protein level of CDX2 was evaluated. The results of western blotting have been added to Figure 4c, and some of the data in previous Figure 2d was added to Figure 4b to avoid confusion. According to your suggestions, we have also added a result of the proliferation assay in miR-9-5p transient-expressed SW480 cells.

Reviewer 4 Report

Comments to Editor

The authors described miRNA-9-5p as a regulator of CDX2 expression, and high miRNA-9-5p expression group (TCGA database) and CDX2 negative group (immunohistochemistry) were associated with poor prognosis in stage â…¡/â…¢ colorectal cancer patients. The finding of the correlation between miRNA-9-5p and CDX2 is interesting. However, the authors could address critical points described below to further improve this article.

Major Point

Authors should describe the mechanism of the relationship between CDX2 expression and the therapeutic effect of NAC/NACRT.

Minor points

The title “A lack of CDX2is associated with a subgroup of patients with stage â…¡/â…¢ CRC(Results, page 2, line 65)” is unclear. Authors should not repeat the result in “Result” section and “Figure legends” to avoid the redundancy. Authors should explain the “miR-9-1” (Results, page 5, line 117) to understand easily. Overexpression of miR-9-1 increased the sensitivity to 5-FU and CPT-11 significantly (Results, page 6, line 139-141)? Please add the P-value in Fig.3c.

Author Response

Response: Thank you for your useful suggestions.

Regarding the “Major Point”, a previous report described the relationship between CDX2 expression and the therapeutic effect of chemotherapies. We have added this information to the “Discussion” section (L239-244).

-Minor Points-

Based on your suggestions, we have modified the title. In addition, the contents of the “Materials and Methods” and “Results” sections have been confirmed, and the Figures have been edited for clarity. We apologize for our redundant description.

The mature miR-9 transcript is produced by three independent genes: miR-9-1, miR-9-2, and miR-9-3. Of miR-9-5p and miR-9-3p, which are mature miR-9s, miR-9-5p is dominant in humans. We used the CRC cell line overexpressing miR-9-1, which stably expresses miR-9-5p, for our basic experiments (Fig4).

Round 2

Reviewer 2 Report

The authors have addressed my concerns.

Reviewer 3 Report

The authors answered questions, made additional experiments and modified their manuscript. I have nothing more to add.

Reviewer 4 Report

Since the authors changedthe manuscript, revised manuscript should be published.